# Role of Cardiac Magnetic Resonance Imaging in the Evaluation of MINOCA

**DOI:** 10.3390/jcm12052017

**Published:** 2023-03-03

**Authors:** Justin A. Daneshrad, Karen Ordovas, Lilia M. Sierra-Galan, Allison G. Hays, Mamas A. Mamas, Chiara Bucciarelli-Ducci, Purvi Parwani

**Affiliations:** 1Department of Internal Medicine, Loma Linda University Health, Loma Linda, CA 92354, USA; 2Department of Cardiothoracic Imaging, University of Washington, Seattle, WA 98195, USA; 3The American British Cowdray Medical Center, Mexico City 05348, Mexico; 4Division of Cardiology, School of Medicine, Johns Hopkins University, Baltimore, MD 21218, USA; 5Keele Cardiac Research Group, Institutes of Science and Technology in Medicine and Primary Care, Keele University, Stoke-on-Trent, Staffordshire ST4 2DE, UK; 6Royal Brompton and Harefield Hospitals, Guys’ and St Thomas NHS Foundation Trust, London SE1 7EH, UK; 7School of Biomedical Engineering and Imaging Sciences, Faculty of Life Sciences and Medicine, Kings College London, London WC2R 2LS, UK

**Keywords:** myocardial infarction with non-obstructive coronary arteries, MINOCA, acute myocardial infarction, coronary artery disease, angiography, Cardiac magnetic Resonance Imaging, myocardium

## Abstract

Myocardial infarction with Non Obstructive Coronary Arteries (MINOCA) is defined by patients presenting with signs and symptoms similar to acute myocardial infarction, but are found to have non-obstructive coronary arteries angiography. What was once considered a benign phenomenon, MINOCA has been proven to carry with it significant morbidity and worse mortality when compared to the general population. As the awareness for MINOCA has increased, guidelines have focused on this unique situation. Cardiac magnetic resonance (CMR) has proven to be an essential first step in the diagnosis of patients with suspected MINOCA. CMR has also been shown to be crucial when differentiating between MINOCA like presentations such as myocarditis, takotsubo and other forms of cardiomyopathy. The following review focuses on demographics of patients with MINOCA, their unique clinical presentation as well as the role of CMR in the evaluation of MINOCA.

## 1. Introduction

Myocardial Infarction with Non-Obstructive Coronary Arteries (MINOCA) refers to the phenomenon of patients presenting with signs and symptoms of acute myocardial infarction with no evidence of significant obstructive coronary artery disease (CAD) on angiography [1]. Beginning with the first pathological report of a patient with acute MI and non-obstructive CAD, our understanding and definition of MINOCA have evolved, culminating in the most recent definition set forth by the European Society of cardiology in 2017 [2] and the inclusion of the MINOCA section in their current guidelines. Although initially, MINOCA was considered to have a relatively benign course, evidence now suggests that patients with MINOCA carry a worse prognosis when compared with the general population [3]. These findings and the observed higher incidence have fueled further research to standardize the pathway to MINOCA diagnosis ultimately to optimize treatment. In recent guideline statements by both the European Society of Cardiology (ESC) and American Heart Association (AHA), Cardiac Magnetic Resonance Imaging (CMR) has become a strong recommendation (class 1B) when evaluating patients with suspected MINOCA [4]. This development is due to diagnostic consistency, accurate assessment of the myocardium, and the minimally invasive nature of CMR.

As CMR has become more readily accessible, imaging-based characteristics of myocardial disease have become increasingly well defined. These imaging characteristics can be used in practice to further differentiate etiologies of myocardial damage which may have similar presentations. Acute myocardial infarction, myocarditis, stress induced cardiomyopathy and MINOCA all share potential overlap regarding clinical stigmata as well as laboratory evidence. In this context the role of CMR becomes all the more impactful, allowing for the use of different imaging techniques to provide tangible evidence in support of a single pathology.

This article discusses the background of MINOCA evaluation leading up to current guidelines and recommendations, ultimately culminating in the unique and pivotal role of CMR within the pathway of MINOCA diagnosis.

## 2. Evolution of Definition of Myocardial Infarction and Prevalence of MINOCA

To understand the fundamental diagnostic difference between MINOCA and acute myocardial infarction (MI), one must consider the evolution in the diagnostic criteria of MI. Beginning in the year 2000, a new definition of acute MI was put forth by the ESC and the American College of Cardiology (ACC) guideline Committee. The redefinition of MI required the rise and fall of biochemical markers such as troponin or CK-MB in addition to one of the following: ischemic symptoms, development of pathologic Q waves on electrocardiogram (ECG), ECG changes consistent with ischemia such as ST-segment elevation or depression, or coronary artery intervention via angiography [5]. The initial definition was refined eventually to the third universal definition of MI, which required elevation of cardiac biomarkers, typically a cardiac troponin >99th percentile of the upper reference level with typical rise and fall in addition to evidence of infarction which includes: typical chest pain, ischemic changes on ECG, and regional wall motion abnormalities appreciated on echocardiography [6]. However, this definition does not adequately differentiate between ischemic and non-ischemic causes of cardiac injury. Some non-ischemic causes may also potentially lead to elevations in troponin values, including myocarditis, stress induced cardiomyopathy (takotsubo syndrome), as well as other non-ischemic cardiomyopathies. Additionally, elevations in troponin are also seen with noncardiac etiologies, such as pulmonary embolism.

Given these limitations of the third universal definition of MI, there was an additional revision made in the fourth universal definition of MI, where the ESC redefined the concept of myocardial injury to provide a more comprehensive assessment. Although both myocardial infarction and myocardial injury involve a significant rise in serum troponin above the 99th percentile, this rise, when seen in myocardial injury, is due to nonischemic injury to the myocytes (as in the case of myocarditis, for example). In contrast, with myocardial infarction, the primary pathology of myocardial injury is due to ischemia [7]. To provide further clarity regarding causes of troponin elevation, Troponin positive with Non-Obstructive Coronary Arteries (TpNOCA) serves as an umbrella term for various pathologies which may result in troponin elevation with non-obstructive coronary arteries. Such pathologies may include ischemic necrosis, structural myocardial disorders and extracardiac disorders [8]. Etiologies of TpNOCA resulting from structural myocardial causes include cardiomyopathies, tachyarrhythmias, cardiotoxicity, myocarditis, and others. Comparatively, non-cardiac etiologies of TpNOCA include pulmonary embolism, renal dysfunction, aortic dissection, etc. [9].

Of the causes of TpNOCA, MINOCA is differentiated by myocardial injury specifically caused by ischemic damage. MINOCA stems as a form of MI where there is diagnostic evidence of ischemia via troponin elevation, ECG changes, and clinical presentation consistent with MI; however, there must be no evidence of obstructive disease, which is defined by less than 50% stenosis identified on coronary angiography. MINOCA is identified in approximately 4–13% of patients who have presented with acute ST-elevation myocardial infarction (STEMI) and have subsequently undergone coronary angiography [10,11]. Data from large acute MI studies show a prevalence of MINOCA between 2–10% [11,12]. Furthermore, when analyzing clinical characteristics in patients presenting with ACS who are found to have non-obstructive disease, MINOCA is overrepresented in women and younger patients. In fact, the incidence of MINOCA among the general population is nearly twice as common in women as in men [12,13]. However, a recent large study demonstrated that there is no difference in all-cause mortality between sexes in patients presenting with suspected MINOCA [14].

Although outcomes of patients with MINOCA are favorable compared to patients with obstructive MI, the overall prognosis is not as benign as once thought. A study by Kang et al. showed that patients with MINOCA had similar prognoses and outcomes compared to patients with one to two vessel coronary artery disease [14,15]. In patients presenting with acute coronary syndrome (ACS) and found to have non-obstructive disease, although the baseline cardiovascular risk is lower, they are still at high risk for cardiovascular morbidity and mortality with yearly major event rates up to 9% [16] comparable to one vessel CAD MI. These findings are further confirmed by multiple systemic reviews [17,18], which show an increased risk of major adverse cardiac events (MACE) in patients presenting with MINOCA on long-term follow-up.

## 3. Path to MINOCA Diagnosis

Considering increasing awareness as to the significance of MINOCA, a diagnostic algorithm aimed at standardizing the diagnostic pathway was put forth by Tamis-Holland et al. They described a stepwise “traffic light” approach to MINOCA diagnosis [19]. The algorithm starts with careful clinical evaluation and diagnostic exclusion of overt causes of myocardial injury like sepsis, pulmonary embolism, etc. Next, it is recommended to review angiographic findings for overlooked angiographic diagnoses. Finally, there must be careful review and consideration for further investigation to rule out non-ischemic causes of myocardial injuries, such as cardiomyopathies, tachyarrhythmias, cardiotoxicities, pulmonary embolism, etc. Even though LV function assessment is commonly performed via transthoracic echocardiography (TTE), it is at this stage that CMR imaging is introduced in the algorithm since it provides tissue characterization, can exclude non-coronary causes of cardiac injury, and can confirm the presence of myocardial injury due to ischemia with susceptible areas of involvement.

Based on a meta-analysis of studies using CMR as a diagnostic tool in patients presenting with a suspected diagnosis of MINOCA, myocarditis was shown to be the leading final diagnosis, identified in up to 38% of cases [17,18,20]. Other major identified causes include Takotsubo cardiomyopathy acute MI, and hypertrophic or dilated cardiomyopathy. After non-ischemic causes are excluded; the possible mechanisms for MINOCA include rupture of atherosclerotic plaque, coronary thrombosis and emboli, microvascular disease, coronary spasm, and spontaneous coronary artery dissection (SCAD). Coronary hemodynamic studies and further coronary imaging with optical coherence tomography (OCT) and intravascular ultrasound (IVUS) can be used successfully to evaluate the coronary etiology further [21].

In a systematic review performed by Pasupathy et al. combining 26 studies with around 1500 MINOCA patients, 33% of patients had myocarditis versus 24% who had a myocardial infarction,18% with Takotsubo syndrome (TS), while 12% had other cardiomyopathy diagnoses. No diagnosis was established in around 26% of the patients [3].

## 4. CMR in MINOCA: Timing, Protocol, and Diagnostic Possibilities

CMR is considered an important tool in the diagnostic pathway for patients presenting with a working diagnosis of MINOCA. With its parametric capabilities in evaluating cardiac structure, function, and myocardial tissue, CMR is considered an important diagnostic tool for assessing the etiology of patients presenting with troponin elevation and myocardial injury.

On the path to MINOCA diagnosis, CMR garners with it a unique capability to accurately assess both ischemic and non-ischemic etiologies for myocardial injury. Recent data supports the early use of CMR in such patients to provide the highest and most accurate diagnostic yield [22]. In fact, when compared with CMR provided diagnoses, individual physicians were shown to miss the diagnosis in up to 25% of cases with that number increasing to 75% for acute MI [23]. In clinical practice, even with experienced clinicians at the helm, there is evidence of poor clinical accuracy as well as low inter-physician agreement in the diagnosis of troponin-positive chest pain with unobstructed coronary arteries without the use of CMR [23].

Multiple studies have corroborated the ability of CMR to accurately provide a diagnosis in patients with MINOCA in 30–90% of cases [9,24,25] and in this process accurately differentiate ischemic from non-ischemic causes. The timing of CMR in such cases contributes to such large variability in the diagnostic utility of CMR in cases of suspected MINOCA. Performing CMR within 2 weeks from index presentation is associated with the highest likelihood of identifying the underlying etiology of myocardial injury in these patients [26]. More rapid implementation of CMR within a week after the index event results in higher diagnostic accuracy of ~90% [22,27]. Recent reports by Williams et al. show that the early implementation of CMR, less than 14 days from presentation, improves diagnostic yield from 72% to 94% when associated with a peak troponin elevation >211 ng/L [20]. Even when CMR is used within three months of index presentation, an identifiable basis for troponin elevation was found in 65–75% of cases [22,25,28]. Other important reasons for the observed variability can be the lack of availability of advanced imaging mapping sequences or the exclusion of T2 weighted edema sequence. Although conventional CMR sequences such as cine imaging and late gadolinium enhancement (LGE) can evaluate myocardial structure and fibrosis [28], parametric mapping sequences and edema imaging can identify segmental abnormalities, interstitial fibrosis, and segmental myocardial edema with >90% sensitivity and specificity [29]. These mapping techniques are particularly promising in locating the areas of myocardial injury responsible for the acute presentation of MINOCA, especially in patients with normal ejection fraction without any regional wall motion abnormalities or LGE, such as in cases of transient vasospasm.

Studies have shown that CMR helps with accurate diagnosis and reclassifies the diagnosis made by clinicians in patients presenting with suspected MINOCA [26]. To test this, Pathik et al. compared the initial diagnosis provided by a panel of three cardiologists compared with a diagnosis obtained after CMR was performed. The panel was blinded to each other’s diagnoses and the CMR diagnosis. The study showed concordance between CMR and physician consensus in only 52% of patients, with a poor level of agreement between the consensus panel and CMR *k* = 0.38, *p* < 0.05 [23]. The highest level of disagreement between the consensus panel and CMR was found in acute MI diagnosis. This study shows the limitations of diagnosis solely made by clinical judgment in the patients presenting with a suspected diagnosis of MINOCA. Dastidar et al. studied a cohort of 204 MINOCA patients with an unclear discharge diagnosis sent for CMR <2 weeks or >2 weeks after an index episode of MINOCA. CMR resulted in a new diagnosis in 54% of cases, with a subsequent change in management in 41% [26]. The study also showed higher diagnostic yield and clinical impact in the propensity matched early CMR group. This data shows the significance of early CMR in ensuring rapid and accurate diagnosis, which is crucial to provide adequate treatment and follow-up to improve overall outcomes in such patients.

## 5. Myocarditis

Myocarditis represents the most common differential diagnosis in patients presenting with troponin elevation in cases of suspected MINOCA ranging from 17–38% [18,30]. The diagnosis of myocarditis heavily relies on clinical presentation including symptom evaluation and history, in combination with imaging and other diagnostic modalities. Patients with myocarditis typically present with chest pain, dyspnea, palpitations fatigue or syncope all of which are nonspecific and can be found with many other cardiac pathologies [31]. The main factors differentiating myocarditis from other cardiac etiologies of MINOCA include fevers, flu-like symptoms, and a reported recent history of viral infection [31]. Although troponin elevation is commonly seen, it is important to note that the extent of troponin elevation does not correlate with the severity of myocardial dysfunction in AM cases [32]. ECG abnormalities are found in nearly 85% of myocarditis cases, specifically with ST elevations involving the inferior occurring as the most common form of ECG abnormality [33]. High risk myocarditis may also present with QRS widening >120 ms, atrioventricular block, symptomatic bradycardia or ventricular arrhythmia [33]. Transthoracic Echocardiography (TTE) is the initial imaging modality of choice in the diagnostic pathway for acute myocarditis (AM). TTE is largely relied on due to its widespread accessibility, rapidity of implementation and relative reliance, however TTE has shown to have limited diagnostic accuracy in cases of AM when compared with more advanced imaging such as CMR [34].

Typical echocardiographic findings seen in AM, such as pericardial effusion, segmental hypokinesia, abnormal myocardial echogenicity and increased myocardial thickness may also be found in other etiologies of myocardial dysfunction [34]. It is worth noting that echocardiographic findings in AM evolve rapidly from the time of diagnosis to during and after treatment [35]. Catheter assisted direct myocardial biopsy remains the gold standard for the diagnosis of AM, which is minimally invasive and requires a physical tissue sample of the myocardium. However, CMR has been proven to be the most sensitive and specific noninvasive modality for the diagnosis of AM due to its ability to both identify and quantify the extent of inflammation and fibrosis [36].

Revised 2018 Lake Louise Criteria aims to standardize the diagnostic approach to AM based on CMR findings. The Lake Louise Criteria includes major and supportive criteria for the diagnosis of AM. Major criteria include: (1) T2-based myocardial edema seen by increasing signal intensity on T2 weighted imaging or T2 parametric mapping, (2) Increased signal intensity with early gadolinium enhancement (3) fibrosis based on LGE [37] (Figure 1). As identified by the revised Lake Louise Criteria, supportive criteria include pericardial effusion, abnormal T1, T2, or LGE, and regional or global wall motion abnormalities. Out of the above diagnostic criteria, just one positive is sufficient to support the diagnosis of AM with strong clinical suspicion. With two out of three criteria identified, AM can be diagnosed with 74% sensitivity and 86% specificity [38].

CMR has more recently been shown to play a significant role in the prognostication of patients presenting with myocarditis. In patients following up at six months after diagnosis, evidence of LGE without edema, especially when seen in a mid-wall septal pattern has shown to be associated with a worse prognosis. LGE seen with a lack of edema is suspected to correlate with definite fibrosis and irreversible myocardial damage, whereas the presence of edema may provide a potential for recovery [39].

## 6. Acute MI

Although in most cases, patients presenting with acute MI (AMI) may have obstructive coronary artery disease, CMR continues to play a role in assessing patients with MINOCA, particularly those with borderline or normal findings on TTE. With CMR, clinicians can accurately assess myocardial structure and function and identify areas of segmental edema, acute inflammation, or chronic fibrosis. CMR has also been used in assessment of patients presenting with various complications of reperfusion therapy such as microvascular obstruction and hemorrhage [40]. In combination, these characteristics allow for the differentiation between acute and chronic ischemic myocardial injury [7]. In addition, using LGE, we can also identify specific forms of scarring that can differentiate ischemic and non-ischemic etiologies. For example, the typical pattern of ischemic fibrosis extends from the sub-endocardium to the epicardium, while non-ischemic fibrosis usually involves the epicardium, mid-wall, or insertion points of the right ventricle [7] (Figure 2). Once the diagnosis of MI is established by CMR, the mechanism of MI of can be further evaluated by the use of intravascular imaging like OCT [19].

CMR plays a strong role in risk stratification and long-term monitoring of cardiac function and remodeling in patients with AMI. As the gold standard for accurate visualization and assessment of cardiac structure and function, CMR allows for evaluating myocardial wall thickness and regional wall motion, which can be affected in acute or chronic ischemic injury. In addition, using both T1 mapping and LGE allows for quantification of MI size [41]. As identified by CMR, MI size has been shown to strongly associate with all-cause mortality and hospitalization even after revascularization [42]. Additionally, CMR has a unique role in identifying microvascular ischemia, which appears as a dark hypointense core within areas of hyperenhancement on EGE or LGE (Figure 3) [7]. Severe microvascular obstruction post-STEMI carries a high risk of intramyocardial hemorrhage and significant mortality [43]. As additional studies continue to emerge, CMR has been shown to have a significant benefit in the acute setting as well as with long-term monitoring of AMI.

## 7. Stress-Induced Cardiomyopathy

Stress induced cardiomyopathy, also known as Takotsubo Syndrome (TS) is an acute heart failure syndrome typically seen in the setting of severe neurologic or emotional stress. It is postulated that TS is driven by a neurocardiogenic connection resulting in cardiac stunning in the presence of severe stress [44]. The presentation of patients with TS can be nearly identical to those with acute coronary syndrome; typical presentation includes the presence of chest pain, dyspnea, troponin elevation and left ventricular dysfunction noted on echocardiogram. Most importantly however, in patients presenting with acute TS coronary angiography will show no signs of obstructive disease.

Although generally reversible, it has been noted there can be significant morbidity and mortality associated with takotsubo syndrome [45,46]. Significant subsequent complications have been shown to arise from takotsubo syndrome as well, including persistent heart failure, valvular abnormalities, cardiogenic shock and arrythmia [47]. Additionally, there has been recent evidence to suggest the presence of various levels of non-complicated atherosclerotic plaques and thin capped fibromas identified via intravascular ultrasound (IVUS) in TTS patients, further contributing to long term morbidity [48].

Echocardiography has remained a crucial part in the pathway to TS diagnosis. There is a distinctive left ventricular (LV) dysfunction pattern associated with TS involving the distal portion of the LV, commonly referred to as apical ballooning [49]. The ventricular dysfunction identified with TS generally involves a circumferential area of myocardium that exceeds the distribution of a single coronary vessel making ischemic damage appear less likely. However, in cases of multivessel disease or less severe cases of TS the presentation on echocardiography may be near identical. CMR is uniquely equipped for diagnosing and monitoring TS due to its advanced anatomical visualization and tissue characterization qualities, as well as the accuracy CMR provides with volumetric measurements. Cine sequencing available with CMR is able to accurately assess contractile function and show in detail the degree of apical ballooning typically seen in such cases [50]. More recently, advanced features available through CMR such as tissue tracking provide an objective measurement of tissue strain and wall motion abnormalities. Not only can this provide meaningful differentiation between TS and similar causes of troponin elevation, but it can be used to identify less common patterns of TS such as midventricular ballooning, basal ballooning and biventricular pattern [44].

Additional criteria for TS include T2 weighted assessment of myocardial edema in non-coronary distribution, increased early gadolinium enhancement uptake, and absence of high signal areas in LGE [51]. T2 weighted imaging analysis of the myocardium in TS patients generally shows patterns of edema which can be used to differentiate from other causes of myocardial damage. Diffuse transmural edema is commonly seen in TS patients, a pattern very different to that of myocarditis which shows sub-epicardial or mid-myocardial edema [44]. Additionally, the area of edema involvement seen with TS is not confined by a single vascular territory, comparatively with MI patterns of edema remaining within a single region correlating with the affected vessel.

## 8. CMR to Prognosticate Patients Presenting with Suspected MINOCA

Patients presenting with suspected MINOCA do not have a benign prognosis. In a study conducted by Kang et. al, patients with MINOCA were shown to have similar prognoses and outcomes compared to patients presenting with one to two vessel disease [14,15]. Recent large cohort studies had shown a 15% mortality in patients with suspected MINOCA that were found to have CMR evidence of cardiomyopathy [25]. In comparison, 24% of patients with suspected MINOCA experienced MACE at 7.1 years follow-up when a CMR diagnosis of myocarditis, AMI, or TS was established [18]. In another study, the strongest predictors of mortality were a CMR diagnosis of cardiomyopathy and ST-segment elevation on presentation ECG at the time of index presentation [23]. Detection of LGE on CMR has been associated with MACE [52]. Additionally, LGE extension into more than two segments of transmural LGE was identified as the strongest predictor of major adverse events in MINOCA [52]. Central Illustration of CMR in patients presenting with suspected MINOCA can be concluded by Figure 4.

## 9. Conclusions

As our understanding of MINOCA continues to evolve, CMR has proven to be an essential tool for establishing the diagnosis in patients presenting with suspected MINOCA. In addition, the timely use of CMR can establish the etiology of myocardial injury to help manage and risk stratify these patients. CMR can be implemented with high diagnostic accuracy to delineate various etiologies in patients presenting with non-specific symptoms such as chest pain and dyspnea who are subsequently found to have troponin elevation. The multimodal nature of CMR provides different imaging techniques to identify and differentiate etiologies of myocardial damage such as acute myocardial infarction, myocarditis and stress induced cardiomyopathy. The main limitations with CMR in assessment of MINOCA lies mainly in availability at the time of the index presentation of MI, especially when considering more rural healthcare settings [53]. Although variability in image interpretation was once a significant limiting factor as well, CMR has become increasingly accessible and mainstream resulting in major improvements to interpreter experience and expertise.

## Figures and Tables

**Figure 1 jcm-12-02017-f001:**
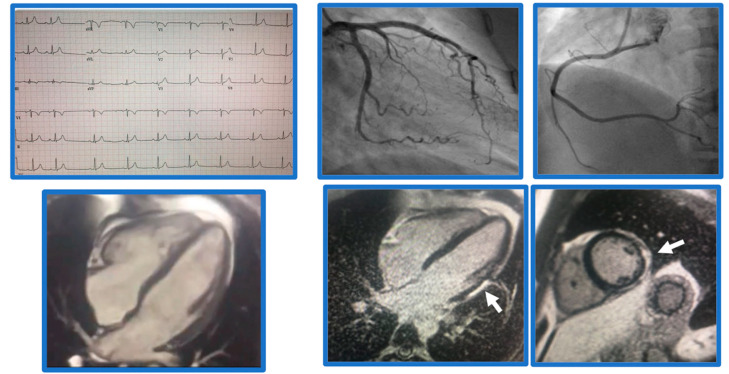
Demonstrative case of myocarditis. Case of a 21-year-old male patient presenting with chest pain. Found to have normal coronaries, SSFP cine imaging showed borderline LV function, EF 54%, and subepicardial basal-mid inferolateral late gadolinium enhancement (arrows), suggestive of myocarditis.

**Figure 2 jcm-12-02017-f002:**
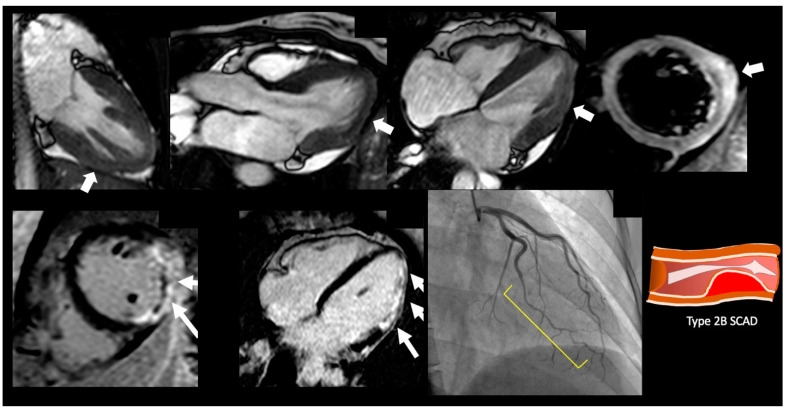
Demonstrative case of a spontaneous coronary artery dissection. A 47-year-old female presenting with chest pain, with nonobstructive coronary artery disease. On SSFP cine imaging showed borderline cardiac function with EF 50% with myocardial edema of the inferolateral wall and corresponding transmural late gadolinium enhancement in inferolateral wall with microvascular obstruction. On cardiac catheterization she was found to have type 2B SCAD lesion of the left circumflex artery.

**Figure 3 jcm-12-02017-f003:**
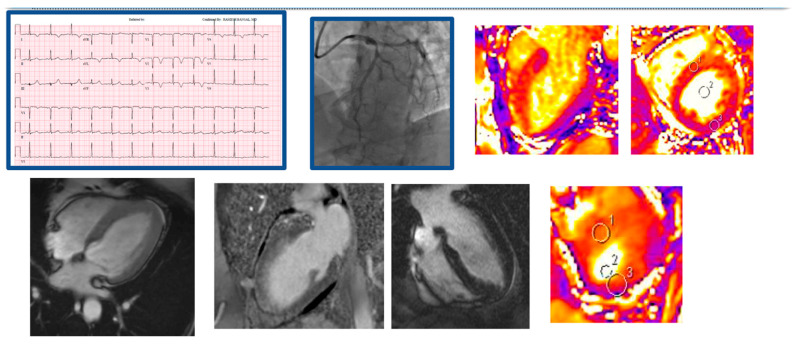
Demonstrative case of a myocardial infarction and acute LAD vasospasm. A 52 yo female with recurrent admissions with chest pain presented anterior STEMI on EKG, troponin I = 0.87, on cardiac catheterization found to have 50% LAD lesion responsive to nitroglycerine, LVEF was found to be 25% on initial TTE. Areas 1–3 suggest areas of interest to measure mapping. The presence of myocardial edema in the LAD territory without any obvious late gadolinium enhancement, suggested myocardial stunning.

**Figure 4 jcm-12-02017-f004:**
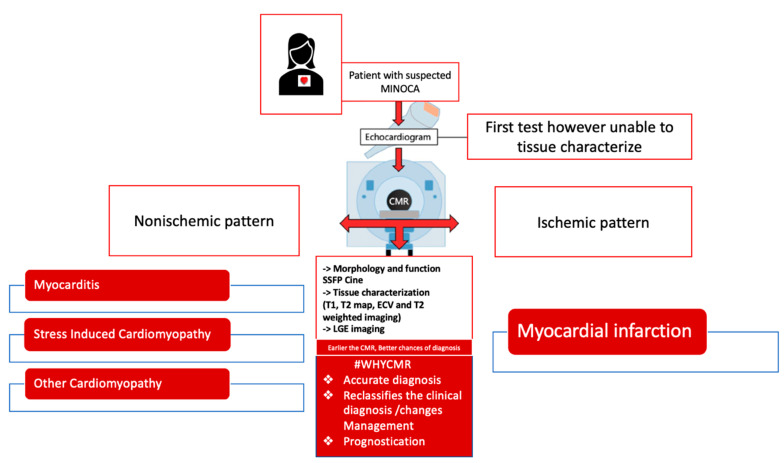
Role of CMR in patients presenting with suspected MINOCA.

## Data Availability

Not applicable.

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
