# Peer review of "Role of Cardiac Magnetic Resonance Imaging in the Evaluation of MINOCA"

_jcm, 2023, doi:10.3390/jcm12052017_

Round 1

Reviewer 1 Report

The authors reviewed MINOCA studies on CMR. They concluded that CMR has become a strong recommendation when evaluating patients with suspected MINOCA ; This development is due to diagnostic consistency, accurate assessment of the myocardium, and the minimally invasive nature of CMR.  

however, it is not clear how distinguish the origins of MINOCA in CMR. Please show the deference of CMR methods and findings for differential diagnosis in patients with MINOCA including diagnosis accuracy.

Author Response

Thank you for your suggestions, the complete revision response has been included the attached document.

Reviewer 2 Report

Daneshrad J A reviewed the role of cardiac magnetic resonance (CMR) imaging in patients with troponin-positive and non-obstructive coronary arteries (TpNOCA).

 This manuscript is well written, however, it should be improving.

 In the manuscript the authors describe the role of (CMR) in patients with ischemic pattern, myocarditis and stess cardiomyopathy. For this reason, I suggest changing the title to: “the role of cardiac magnetic resonance imaging in patients with troponin positive and non-obstructive coronary artery.”

When the authors discuss the possible pathophysiological mechanisms of MINOCA (as well defined by the authors), the mechanism may be microvascular spasm and microvascular disease, if other diseases can get worse in the coronary microvascular circulation.

Please provide some data regarding the prognosis in MINOCA patients (you can cite the following paper: PMCID: PMC5210396 DOI: 10.1161/JAHA.116.004185 and reference number 22). Moreover, the authors should discuss that MINOCA patients exhibited modest myocardial damage (PMCID: PMC6064896  DOI: 10.1161/JAHA.118.009174 ) but the prognosis is not well (please cite: PMCID: PMC8464114 DOI: 10.1186/s12933-021-01384-6)

CMR in TpNOCA should be performed within seven days after acute presentations.

In the subheading myocarditis, the author should also discuss the importance of clinical presentation, and the clinical presentation does not include flu-like symptoms with a positive history of viral infections, a common and frequent occurrence in the myocarditis population (Ammirati E. et al. Circ Heart Fail. 2020 Nov;13(11):e007405. doi: 10.1161/CIRCHEARTFAILURE.120.007405).

The authors should discuss the role of different forms of CMR (ischemic and non-ischemic) in a possible therapeutic program. In ischemic pattern, the role of secondary prevention therapy is not well defined (please cite: PMCID: PMC7005107 DOI: 10.3389/fphar.2019.01606; PMID: 28179398 DOI: 10.1161/CIRCULATIONAHA.116.026336

Reference number 22 needs to be cited well; please provide.

Author Response

(The authors gave the same response as above.)

Reviewer 3 Report

I reviewed with interested this paper by Daneshrad et al. abour the role of CMR in patients with MINOCA. This review article is well written and represents an updated overview on this topic.

I have only minor comments in order to further improve the overall quality of the manuscript.

- Page 3 line 108. Please note that along plaque rupture also plaque erosion may represent a cause of coronary plaque instability (see PMID: 34600948).

- It is not clear to this reviewer how CMR may help to differentiate among different ischemic mechanisms underlying true MINOCA (eg. differentiating epicardial from non-epicardial pattern, etc.; see PMID: 35111833).

- You can consider citing some relevant articles on this topic already published (PMID: 34600948 and 35111833).

_Please, briefly discuss limitations of CMR in this setting.

Author Response

(The authors gave the same response as above.)

Round 2

Reviewer 2 Report

None